# Life span inequality as a function of the moments of the deaths distribution: Connections and insights

**Oscar E. Fernandez**[1]*, **Hiram Beltrán-Sánchez**[2]*

**1** Department of Mathematics, Wellesley College, Wellesley, MA, United States of America, **2** Department of Community Health Sciences, UCLA Fielding School of Public Health, and California Center for Population Research, Los Angeles, California, United States of America

* ofernand@wellesley.edu (OEF); beltrans@ucla.edu (HBS)

**Data Availability Statement:** All data analyzed are from the Human Mortality Database (https://www.mortality.org/).

## Abstract

Recent work has unearthed many empirical regularities in mortality trends, including the inverse correlation between life expectancy and life span inequality, and the compression of mortality into older age ranges. These regularities have furnished important insights into the dynamics of mortality by describing, in demographic terms, how different attributes of the life table deaths distribution interrelate and change over time. However, though *empirical* evidence suggests that the demographically-meaningful metrics these regularities involve (e.g., life span disparity and life table entropy) are correlated to the moments of the deaths distribution (e.g., variance), the broader *theoretical* connections between life span inequality and the moments of the deaths distribution have yet to be elucidated. In this article we establish such connections and leverage them to furnish new insights into mortality dynamics. We prove theoretical results linking life span disparity and life table entropy to the central moments of the deaths distribution, and use these results to empirically link statistical measures of variation of the deaths distribution (e.g., variance, index of dispersion) to life span disparity and life table entropy. We validate these results via empirical analyses using data from the Human Mortality Database and extract from them several new insights into mortality shifting and compression in human populations.

## Introduction

It is a well-documented phenomenon that social health risk factors, (e.g., social determinants of health) and social conditions operate as fundamental causes of disease and death (see, e.g., [1–3] and references therein). These factors, in turn, underlie the variability in ages at death in a population whereby, for example, individuals with low socioeconomic status and/or with social disadvantage tend to die at younger ages than those in more privileged positions [4]. In demography, two of the most well-studied metrics for quantifying such inequalities in life spans are the *life span disparity*, denoted by $e^{\dagger}$ [5–9] and *life table entropy*, denoted by $H$ [10,

**Funding:** O.E.F. is grateful to the John Simon Guggenheim Memorial Foundation for their generous support of this work via the Guggenheim Fellowship O.E.F. received. H.B.S. acknowledges support from grants by the National Institute on Aging (R01AG052030) and the National Institute of Child Health and Human Development (P2C-HD041022) to the California Center for Population Research at UCLA. The funders had no role in study design, data collection and analysis, decision to publish, or preparation of the manuscript.

**Competing interests:** The authors have declared that no competing interests exist.

11], defined as

$$e^{\dagger} = \int_0^{\omega} e(x)f(x)\,dx, \quad H = e^{\dagger}/e_0, \tag{1}$$

where $\omega$ is the maximum life span, and $e(x)$ and $f(x)$ are the remaining life expectancy and deaths distribution, respectively, at age $x$. Recent work has shown that life span disparity ($e^{\dagger}$) measures the average life expectancy lost due to death [7, 8], while life table entropy measures the average life expectancy lost due to death as a percentage of life expectancy at birth [5] (see [9, 12] for brief summaries of additional interpretations of $e^{\dagger}$ and $H$). These properties have led researchers to use life span disparity and life table entropy as indicators of life span variability [13–16].

Now, at the population level, the inequality in life spans is known to vary considerably across countries, species, and time [17], a finding that has proved empirically robust regardless of which of the inter-correlated metrics of life span inequality one looks at [15, 16, 18, 19]. These variations in life span inequality can be traced ultimately to differences in populations' deaths distributions [20]. Researchers have quantified these differences in terms of the individual moments of the deaths distributions, and in recent decades a plethora of empirical regularities in human populations have been found that connect these individual moments to metrics of life span inequality. For example, studies have found positive correlations between the variance in the deaths distribution and life span disparity and life table entropy [18]. Studies have also documented the increasingly left-skewed nature of human populations' deaths distributions [20, 21]—i.e., mortality has become more concentrated in older age ranges—and connected this phenomenon to decreasing life span inequality via so-called mortality compression [22]. These and similar studies suggest that changes in life span inequality are driven by more than just changes in one individual moment of the deaths distribution. Yet, as pointed out by [20], "only four studies so far have taken into account more than one moment of the age-at-death distributions when examining differences in distributions of ages at death across a group of countries" (see [20] for an overview of those studies and their findings). Furthermore, the study including the largest number of moments to date, that of [20], only incorporates two moments, and like all of the aforementioned studies, is empirical—there are no theoretical rationales provided for the particular associations found, and no theoretical explanations provided for the trends and patterns identified. This theoretical foundation, along with the potential associations between life span inequality and the ignored moments, therefore remains unknown and unstudied. More generally, since there are no known analytical expressions connecting life span inequality metrics to the full set of moments of the deaths distribution, we lack a theoretical understanding of how life span disparity and life table entropy depend, in general, on the moments of the deaths distribution.

In this article we take a first step toward establishing these missing connections by building a bridge between well-known statistical and demographic descriptors of mortality. Across two theorems—Theorem 1 and Corollary 1.1—we derive analytical expressions between the full set of moments of the deaths distribution and life span disparity, and also derive a similar expression for the life table entropy. We also generalize our results to the setting where one chooses nonzero age-values against which to measure life span disparity and life table entropy (i.e., considering the deaths distribution starting at any age, not only from birth as it is typically done). Our results take the form of infinite series expansions of $e^{\dagger}$ and $H = e^{\dagger}/e_0$ (and their generalizations to nonzero starting ages) in terms of the central moments of the deaths distribution. We also show how truncating these series after the first two terms produces approximations that suggest a correlation between life span disparity and the variance of the

deaths distribution, as well as between life table entropy and the index of dispersion of the deaths distribution. We validate these correlations via empirical studies of mortality data from the Human Mortality Database [23] (7,340 1-year life tables of males and females, 41 countries; 1751–2020). We conclude with a discussion of the various demographic insights that emerge from our results.

## Materials and methods

Consider an age-structured population with maximum life span $\omega$ and denote by $x$ the age of members in the population (measured in years). The *distribution of ages at death* (*deaths distribution*, for short) in the population, which we will denote by $f(x)$, encodes information about how aggregate mortality is distributed across age in the population. We assume hereafter that this distribution is constructed from life tables. Denoting by $\mu(x)$ the *force of mortality* function and by $s(x)$ the population's *survival function*, the deaths distribution is defined as

$$f(x) = \mu(x)s(x) = -s'(x), \quad 0 \le x \le \omega. \tag{2}$$

(We assume hereafter that all functions introduced are at least differentiable, unless otherwise indicated; Appendix A.1 in S1 Appendix reviews these life table functions and their relationships.) We also denote by $f_a(x)$ the deaths distribution restricted to ages $x \ge a$,

$$f_a(x) = \frac{f(x)}{\displaystyle\int_a^\omega f(x)\,dx} = \frac{f(x)}{s(a)}, \quad 0 \le a < \omega, \quad a \le x \le \omega. \tag{3}$$

(The denominator in (3) ensures that $f_a$ is a probability density function.) We will refer to (3) as the "*a*-truncated deaths distribution," and note that $f(x) = f_0(x)$, that is, the full deaths distribution (2) is the "0-truncated deaths distribution" obtained by setting $a = 0$ in (3).

To quantify the life span inequality associated with each $f_a$ we recall the generalization of $e^\dagger$ to the *a*-truncated setting [24, 25]:

$$e^\dagger(a) = \int_a^\omega e(x)f_a(x)\,dx, \tag{4}$$

where $e(x)$ is the *remaining life expectancy* at age $x$, defined as

$$e(a) = \frac{\displaystyle\int_a^\omega s(x)\,dx}{s(a)}. \tag{5}$$

(We note in passing that $e^\dagger(0) = e^\dagger$ and $H(0) = H$.) We also define the generalization of $H$ to the *a*-truncated setting as

$$H(a) = e^\dagger(a)/e_a, \tag{6}$$

where $e_a$ is the mean value of $f_a(x)$ (defined below in (7)). Our objective now is to relate $e^\dagger(a)$

and $H(a)$ to this mean value and the central moments $m_n(a)$ associated with $f_a$:

$$e_a = \int_a^\omega x f_a(x)\, dx = \frac{\int_a^\omega x f(x)\, dx}{s(a)}, \tag{7}$$

$$m_n(a) = \int_a^\omega (x - e_a)^n f_a(x)\, dx = \frac{\int_a^\omega (x - e_a)^n f(x)\, dx}{s(a)}, \quad n = 0, 1, 2, \ldots, \tag{8}$$

where we used (3) to obtain the second equalities in each line. Since these quantities are less studied than their $a = 0$ counterparts, let us briefly review their meaning and introduce related quantities we will need before continuing on to present our results connecting $e^\dagger(a)$ and $H(a)$ to $e_a$ and $m_n(a)$.

Demographers recognize the $a = 0$ case of (7) as the *mean age at death* in the stationary population, which is also equivalent to the *life expectancy at birth*, $e_0$ (see Appendix A.1 in S1 Appendix for the calculation that (7) yields $e_0$ when $a = 0$). For $a > 0$, $e_a$ is the *conditional mean age at death*, or equivalently, the mean life span of individuals in the population that have already survived to age $a$. This can be seen by substituting $f(x) = -s'(x)$ (from (2)) into (7) and using integration by parts:

$$e_a = -\frac{\int_a^\omega x s'(x)\, dx}{s(a)} = \frac{a s(a) + \int_a^\omega s(x)\, dx}{s(a)} = a + e(a). \tag{9}$$

Moving on to (8), since $m_0(a) = 1$ and $m_1(a) = 0$, only the $n \geq 2$ central moments provide useful information about $f_a$. In particular,

$$m_2(a) = \int_a^\omega (x - e_a)^2 f_a(x)\, dx =: \sigma^2(a) \tag{10}$$

is the *variance* of the $a$-truncated deaths distribution $f_a$, i.e., the conditional variance in the age at death among those who have survived to age $a$. An alternate but equivalent expression for $\sigma^2(a)$ is given in [26] as

$$\sigma^2(a) = \int_a^\omega (x - a)^2 f_a(x)\, dx - [e(a)]^2. \tag{11}$$

(We note that $\sigma^2(0) =: \sigma^2$, the variance of the distribution $f$.) Following (6), we define the *index of dispersion* of the $a$-truncated deaths distribution $f_a$ as:

$$v_d(a) = \frac{\sigma^2(a)}{e_a}. \tag{12}$$

(We note that when $a = 0$ we obtain $v_d := v_d(0) = \sigma^2/e_0$, the standard definition for the index of dispersion).

Let us now return to the task of relating $e^\dagger(a)$ and $H(a)$ to $e_a$ and the $m_n(a)$. Our main theoretical results—the two theorems below—accomplish this by showing that, under suitable conditions, both $e^\dagger(a)$ and $H(a)$ can be expanded in infinite series whose terms depend on the moments $m_n(a)$. (See Appendix A.2 in S1 Appendix for the proofs of these theorems).

**Theorem 1** *Consider a natural population with maximum life span $\omega$ and remaining life expectancy function $e(x)$, where $x \in [0, \omega]$, and let $a \in [0, \omega)$ and $k$ be a natural number such that $k \geq 3$. If $e(x)$ is analytic on $[0, \omega]$ and $f_a(x)$ is continuous on $[a, \omega]$, then there exist $\xi \in (a,$*

*ω) such that*:

$$e^{\dagger}(a) \quad = \quad e(e_a) + \frac{e''(e_a)}{2!}\sigma^2(a) + \sum_{n=3}^{k} \frac{e^{(n)}(e_a)}{n!} m_n(a) + \frac{e^{(k+1)}(\xi_1)}{(k+1)!} m_{k+1}(a), \tag{13}$$

$$H(a) \quad = \quad \frac{e(e_a)}{e_a} + \frac{e''(e_a)}{2!}v_d(a) + \sum_{n=3}^{k} \frac{e^{(n)}(e_a)}{n!}\frac{m_n(a)}{e_a} + \frac{e^{(k+1)}(\xi)}{(k+1)!}\frac{m_{k+1}(a)}{e_a}. \tag{14}$$

(*Here* $e^{(n)}(e_a)$ *denotes the n-th derivative of e(x) evaluated at $e_a$.*) *Furthermore, the last term in each equation tends to zero as* $k \to \infty$.

**Corollary 1.1** *Under the assumptions of Theorem 1, the a = 0 cases of* (13) *and* (14) *become*:

$$e^{\dagger} \quad = \quad e(e_0) + \frac{e''(e_0)}{2!}\sigma^2 + \sum_{n=3}^{k} \frac{e^{(n)}(e_0)}{n!} m_n(0) + \frac{e^{(k+1)}(\xi_1)}{(k+1)!} m_{k+1}(0), \tag{15}$$

$$H \quad = \quad \frac{e(e_0)}{e_0} + \frac{e''(e_0)}{2!}v_d + \sum_{n=3}^{k} \frac{e^{(n)}(e_0)}{n!}\frac{m_n(0)}{e_0} + \frac{e^{(k+1)}(\xi_2)}{(k+1)!}\frac{m_{k+1}(0)}{e_0}. \tag{16}$$

Theorem 1 and Corollary 1.1 furnish the analytical expressions we noted in the Introduction were missing from the literature—they connect life span disparity and life table entropy, two widely used metrics of life span inequality in demography developed with specific demographic applications in mind, to the full set of moments of the deaths distribution. In what follows we discuss the many insights and ramifications of Theorem 1 and Corollary 1.1. But first let us briefly illustrate these results using a theoretical model of mortality: the *hyperbolic mortality model*.

## Example: The hyperbolic mortality model

In this model,

$$\mu(x) = \frac{k}{\omega - x}, \quad s(x) = \left(1 - \frac{x}{\omega}\right)^k, \quad f(x) = \frac{k}{\omega}\left(1 - \frac{x}{\omega}\right)^{k-1}, \quad k > 0. \tag{17}$$

In Appendix A.3 in S1 Appendix we show that:

$$e_0 = \frac{\omega}{k+1}, \quad e(x) = \frac{\omega}{k+1}\left(1 - \frac{x}{\omega}\right), \quad e^{\dagger} = \frac{k\omega}{(k+1)^2}, \quad H = \frac{k}{k+1}. \tag{18}$$

Since $e(x)$ here is a linear function it is analytic for all $x$ (i.e., for all age values). Moreover, since $f(x)$ is a power function with non-negative exponent for $k \geq 1$ then $f$ is continuous on $[0, \omega]$ for $k \geq 1$. Thus, for $k \geq 1$ both Theorem 1 and Corollary 1.1 apply. We now illustrate the latter. To do so we note that

$$e(e_0) \quad = \quad \frac{\omega}{k+1}\left(1 - \frac{e_0}{\omega}\right) = \frac{\omega}{k+1}\left(1 - \frac{1}{\omega}\left[\frac{\omega}{k+1}\right]\right) = \frac{k\omega}{(k+1)^2},$$

$$e'(e_0) \quad = \quad -\frac{1}{k+1},$$

and $e^{(n)}(e_0) = 0$ for all $n \geq 2$. Thus, (15) and (16) collapse to just their first terms, and we have:

$$e^{\dagger} = e(e_0) = \frac{k\omega}{(k+1)^2}, \quad H = \frac{e(e_0)}{e_0} = \frac{k\omega}{(k+1)^2}\left(\frac{k+1}{\omega}\right) = \frac{k}{k+1}, \tag{19}$$

giving back the last two equations in (18) calculated directly from (1). (We note that these results also hold for $0 < k < 1$, even though $f(x) \to \infty$ as $x \to \omega^-$.) These calculations show that in the hyperbolic mortality model, life span disparity is exactly equal to the remaining life expectancy at age $x = e_0$ (the mean life span at birth), and that life table entropy is the particular ratio of life expectancies $e(e_0)/e_0$.

## Truncating the $e^\dagger(a)$ and $H(a)$ series

We mentioned in the Introduction that to date studies have linked life span inequality to at most two moments of the deaths distribution, and that these have been empirical studies lacking the theoretical rationale to justify or explain the associations found. Eqs (13)–(16) remedy these issues. By truncating those series at term $n = k$ (which is equivalent to disregarding the (last) $m_{k+1}(a)$ term in each equation) one obtains approximations to $e^\dagger(a)$ and $H(a)$ that involve all nonzero central moments up to $m_k(a)$. The resulting truncated series therefore provide theoretical foundations for probing how multiple moments of the deaths distribution interact to drive life span inequality. Furthermore, since the coefficients of the central moments in the truncated series are all related to the remaining life expectancy function and its higher-order derivatives, empirical studies based on those approximations—e.g., regressions—will yield results interpretable in terms of the remaining life expectancy function.

Motivated by these insights, let us therefore truncate the series (13)–(16) to retain only the first four terms, which include the three most studied moments of a distribution (the variance, and the unstandardized skewness and kurtosis):

$$e^\dagger(a) \quad \approx \quad e(e_a) + \frac{e''(e_a)}{2!}\sigma^2(a) + \frac{e^{(3)}(e_a)}{3!}m_3(a) + \frac{e^{(4)}(e_a)}{4!}m_4(a), \tag{20}$$

$$e^\dagger \quad \approx \quad e(e_0) + \frac{e''(e_0)}{2!}\sigma^2 + \frac{e^{(3)}(e_0)}{3!}m_3(0) + \frac{e^{(4)}(e_0)}{4!}m_4(0), \tag{21}$$

$$H(a) \quad \approx \quad \frac{e(e_a)}{e_a} + \frac{e''(e_a)}{2!}v_d(a) + \frac{e^{(3)}(e_a)}{3!}\widehat{m}_3(a) + \frac{e^{(4)}(e_a)}{4!}\widehat{m}_4(a), \tag{22}$$

$$H \quad \approx \quad \frac{e(e_0)}{e_0} + \frac{e''(e_0)}{2!}v_d + \frac{e^{(3)}(e_0)}{3!}\widehat{m}_3(0) + \frac{e^{(4)}(e_0)}{4!}\widehat{m}_4(0), \tag{23}$$

where $\widehat{m}_3(a) = m_3(a)/e_a$, $\widehat{m}_4(a) = m_4(a)/e_a$, and $e^{(n)}(x)$ is the $n$-th derivative of $e(x)$. These approximations connect the life span disparity and life table entropy values associated with a particular life table (also, deaths distribution) to the first three nonzero central moments of the deaths distribution defined by that life table in the $a$-truncated or full age-range cases.

If we now envision applying (20)–(23) to two temporally consecutive life tables—e.g., one at time $t$ and the other at time $t + 1$ (we assume, for simplicity, that $t$ is measured in years)—then (20)–(23) suggest that there may be multilinear associations between life span disparity and life table entropy and the first three nonzero central moments of the population's deaths distribution, with the slopes of these associations depending on the higher-order derivatives of the remaining life expectancy function evaluated at $e_a$ or $e_0$. For this to hold, in the case of the full distribution for example, $e(e_0)$ and $e^{(n)}(e_0)$ for $n = 2, 3, 4$ would need to change much slower than the moments in (20) and (21), and $e(e_0)/e_0$ and $e^{(n)}(e_0)/e_0$ for $n = 2, 3, 4$ would need to change much slower than the moments in (22) and (23). Similar assumptions would need to hold in the $a$-truncated case. In the next section we investigate these possibilities

empirically and present empirical support for approximations (20)–(23) for countries in the Human Mortality Database (HMD) [23].

## Results

### Empirical analyses of Theorem 1 and Corollary 1.1

We now examine Theorem 1 and Corollary 1.1 empirically using data from the Human Mortality Database [23]. This data set contains 7,340 1-year life tables of males and females (41 countries; 1751–2020, see Table 9 in S1 Appendix). Now, Theorem 1 and Corollary 1.1 assume that $e(x)$ is analytic on $[0, \omega]$ and that $f_a(x)$ is bounded on $[a, \omega]$. We were able to verify these assumptions in the hyperbolic mortality case (see the Example section above) because we had the explicit $e(x)$ and $f(x)$ formulas. We do not have such expressions for the HMD data set though. But the analyticity of each country's $e(x)$ functions can nonetheless be established by interpolating the discrete set of $e(x)$-values provided in the life tables using an analytic function, thereby producing analytic $e(x)$ functions for each life table. And because calculating the terms in (13)–(16) does not involve the interpolated values (see Appendix B.1 in S1 Appendix for the discrete formulas employed to calculate the terms in (13)–(16)), the $e(x)$ analyticity assumption is satisfied by construction. Likewise, the boundedness assumption on $f_a(x)$ is satisfied by construction by interpolating the life table deaths distributions using continuous functions. We are thus justified in applying Theorem 1 and Corollary 1.1 to the data set. With this in mind we now turn to Figs 1 and 2, which investigate (20)–(23).

Let us first examine the second-order truncation (that is, the truncation up to $m_2(a) = \sigma^2(a)$) of the approximations (20)–(23):

$$e^\dagger(a) \approx e(e_a) + \frac{e''(e_a)}{2!}\sigma^2(a), \ e^\dagger \approx e(e_0) + \frac{e''(e_0)}{2!}\sigma^2, \tag{24}$$

$$H(a) \approx \frac{e(e_a)}{e_a} + \frac{e''(e_a)}{2!}v_d(a), \ H \approx \frac{e(e_0)}{e_0} + \frac{e''(e_0)}{2!}v_d. \tag{25}$$

(We refer the reader to Appendix B.1 in S1 Appendix for details of our data analysis methods; all analyses were performed using the statistical software R [27].) Figs 1 (top) and 2 (top) investigate the approximations in (24) and (25), respectively, for $a = 0, 10, 40,$ and $80$ for females in the HMD (see Figs 3 and 4 in Appendix B.4 of S1 Appendix for the corresponding figures for males) while Figs 1 (bottom) and 2 (bottom) do the same but only for French females in the data set. We will use the French population to discuss aspects of our theory that cannot be illustrated at the aggregated scale of the 41 countries in the HMD. We note first that for all but $a = 80$, both $e^\dagger(a)$ and $H(a)$ have in general been decreasing since 1751, reflecting an overall trend toward decreasing life span inequality regardless of the starting value ($a$-value) used to measure life span disparity or life table entropy. This is not true of the trends for age 80, however. As the last subfigure in each set of figures shows, $e^\dagger(80)$ and $H(80)$ were generally lower in 1751–1949 than they have been since 1950. That is, conditional on surviving to age 80, there has been an increase in the uncertainty in ages at death above age 80 in recent times. This reflects the ongoing trend of HMD countries' populations reaching and dying at increasingly older ages than age 80. This phenomenon naturally increases $e^\dagger(80)$ and $H(80)$ as it adds progressively more life table deaths at progressively older ages than age 80, thus increasing the dispersion in mortality for the portion of the deaths distribution lying beyond age 80 (this pattern can also be seen in the larger variance and index of dispersion in ages at death above 80).

Turning now to the potential linearity of (24) and (25) we discussed in the previous section, Figs 1 and 2 show that $e^\dagger(a)$ and $H(a)$ both exhibit strong linearity with respect to $\sigma^2(a)$ and

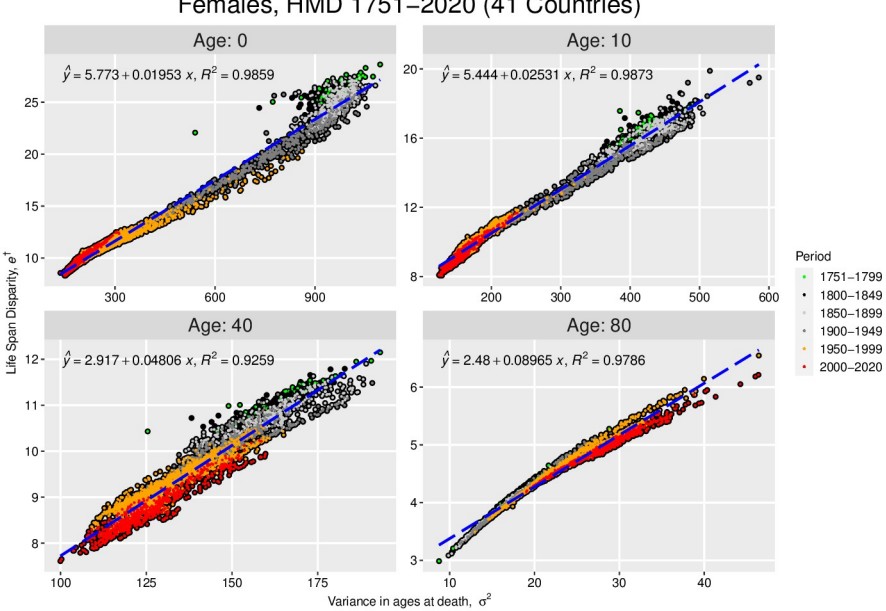

**Fig 1. Life span disparity versus variance in age at death for females in the Human Mortality Database and French National Population.** Plots of the $a$-truncated life span disparity, $e^\dagger(a)$, versus the $a$-truncated variance in age at death, $\sigma^2(a)$, for females in 41 countries in the Human Mortality database [23] (3,670 1-year life tables; 1751–2020) (top), and females in France, National Population from the Human Mortality database [23] (203 1-year life tables; 1816–2018) (bottom). The dashed lines are the plots of the best-fit regression lines; the error statistics associated with the top figure are detailed in Table 5 in S1 Appendix, and those associated with the bottom figure are detailed in Table 1.

$v_d(a)$, respectively, for all the $a$-values studied. Indeed, the associated standard errors and adjusted $R^2$-values (see Tables 1 and 2) confirm the robustness of those linear relations. (These linear relations are weakest at $a = 40$, however, a phenomenon also characteristic of the male populations; see Tables 3 and 4 and Figs 3 and 4 in Appendix B.4 in S1 Appendix) The weak

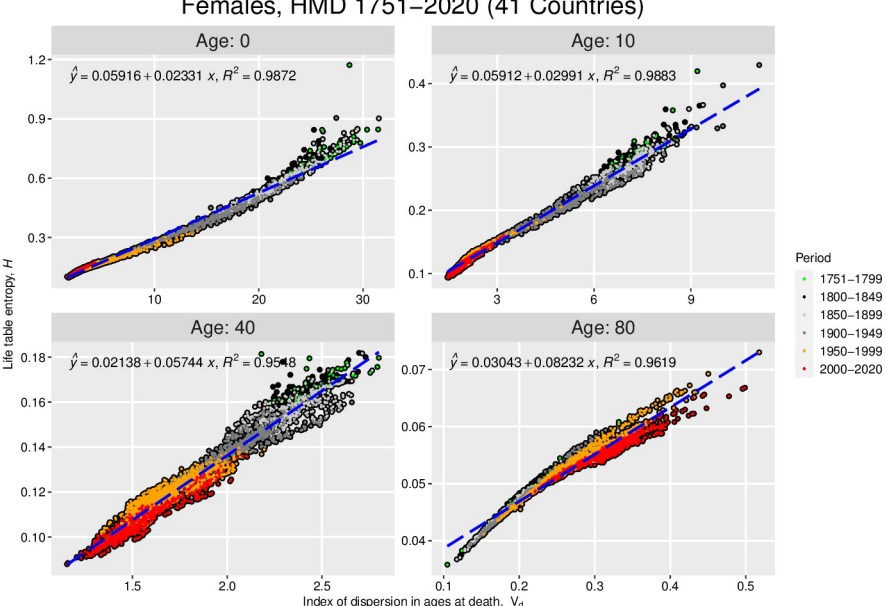

**Fig 2. Life table entropy versus index of dispersion of deaths distribution for females in the Human Mortality Database and French National Population.** Plots of the *a*-truncated life table entropy, *H*(*a*), versus the *a*-truncated index of dispersion of age at death, $v_d(a)$, for females in 41 countries in the Human Mortality database [23] (3,670 1-year life tables; 1751–2020) (top), and females in France, National Population from the Human Mortality database [23] (203 1-year life tables; 1816–2018) (bottom). The dashed lines are the plots of the best-fit regression lines; the error statistics associated with the top figure are detailed in Table 7 in S1 Appendix, and those associated with the bottom figure are detailed in Table 2.

**Table 1. Regression coefficients for life span inequality $e^\dagger$, based on (20) and (21), for French females, National Population (1816–2018) in the Human Mortality Database [23], along with adjusted $R^2$-values and standard errors (S.E.).**

| Parameter | $e^\dagger(0)$ | $e^\dagger(0)$ | $e^\dagger(0)$ | $e^\dagger(40)$ | $e^\dagger(40)$ | $e^\dagger(40)$ | $e^\dagger(80)$ | $e^\dagger(80)$ | $e^\dagger(80)$ |
|---|---|---|---|---|---|---|---|---|---|
| Intercept | 5.10015 | 6.35912 | 6.24003 | 0.53677 | 5.17485 | 4.98710 | 2.49746 | 2.18393 | 1.98932 |
|  | (0.126384) | (0.051246) | (0.04781) | (0.139561) | (0.049281) | (0.047662) | (0.018899) | (0.009671) | (0.00952) |
| $\sigma^2$ | 0.02087 | 0.02066 | 0.01902 | 0.06398 | 0.03980 | 0.04191 | 0.08749 | 0.09595 | 0.11704 |
|  | (0.000169) | (0.000055) | (0.000219) | (0.000932) | (0.000269) | (0.000338) | (0.000949) | (0.000367) | (0.000902) |
| $m_3$ |  | 0.00011 | 0.00015 |  | 0.00086 | 0.00075 |  | 0.00427 | 0.00525 |
|  |  | (0.000003) | (0.000006) |  | (0.000008) | (0.000015) |  | (0.000102) | (0.000066) |
| $m_4$ |  |  | 0.000001 |  |  | -0.000004 |  |  | -0.000206 |
|  |  |  | (0.0000002) |  |  | (0.0000005) |  |  | (0.000009) |
| Adjusted $R^2$ | 0.9869 | 0.9986 | 0.9989 | 0.9589 | 0.9992 | 0.9994 | 0.9768 | 0.9976 | 0.9994 |
| Regression S.E. | 0.7574 | 0.2465 | 0.2174 | 0.1830 | 0.0252 | 0.0216 | 0.0803 | 0.0259 | 0.0132 |

Source: Authors' calculations using data from the Human Mortality Database [23].

All coefficient $p$-values were less than $1 \times 10^{-6}$.

linear association between $e^\dagger(40)$ and $H(40)$ with respect to $\sigma^2(40)$ and $v_d(40)$, respectively, is indicative of a more general pattern that occurs in middle-adult ages given that at these ages the shape of the distribution in ages at death also plays a role. We describe this in the next page when we incorporate more central moments into the regression equations.

The robust relations shown in Figs 1 and 2 make it possible to estimate the associated change in life span inequality resulting from changes in the underlying $a$-truncated variance and index of dispersion values in the population. For example, we expect a decrease of 1 unit in the variance and a decrease of 1 unit in the index of dispersion of the population's death distribution to result in decreases of life span disparity and life table entropy, respectively, of approximately 0.02 units, as indicated by the least-squares regression equations in the top-left corner of the Age 0 subfigures. We noted in the Introduction how previous studies uncovering empirically-derived associations like these have been unable to explain their origins or demographic drivers. However, in our case (24) provides this information. First, we note that the coefficient of $\sigma^2(a)$ and $v_d(a)$ in (24) and (25) is the same for all $a$-values: $e''(e_0)/2$. This helps

**Table 2. Regression coefficients for life table entropy $H$, based on (22) and (23), for French females, National Population (1816–2018) in the Human Mortality Database [23], along with adjusted $R^2$-values and standard errors (S.E.).**

| Parameter | $H(0)$ | $H(0)$ | $H(0)$ | $H(40)$ | $H(40)$ | $H(40)$ | $H(80)$ | $H(80)$ | $H(80)$ |
|---|---|---|---|---|---|---|---|---|---|
| Intercept | 0.04613 | 0.07471 | 0.07039 | 0.00187 | 0.05627 | 0.05646 | 0.03098 | 0.02672 | 0.02433 |
|  | (0.002664) | (0.001462) | (0.001421) | (0.001091) | (0.002351) | (0.000817) | (0.020321) | (0.000115) | (0.000115) |
| $\sigma^2$ | 0.02443 | 0.02433 | 0.02118 | 0.06669 | 0.05065 | 0.05733 | 0.07860 | 0.08882 | 0.11054 |
|  | (0.000161) | (0.000068) | (0.000428) | (0.00052) | (0.000724) | (0.000306) | (0.001187) | (0.00039) | (0.000927) |
| $\widehat{m_3}$ |  | 0.00015 | 0.00021 |  | 0.00132 | 0.00084 |  | 0.00482 | 0.00578 |
|  |  | (0.000005) | (0.000009) |  | (0.000055) | (0.000023) |  | (0.000098) | (0.000064) |
| $\widehat{m_4}$ |  |  | 0.000003 |  |  | -0.000024 |  |  | -0.000206 |
|  |  |  | (0.0000003) |  |  | (0.0000006) |  |  | (0.000009) |
| Adjusted $R^2$ | 0.9913 | 0.9985 | 0.9988 | 0.9879 | 0.9968 | 0.9996 | 0.9560 | 0.9966 | 0.9991 |
| Regression S.E. | 0.0199 | 0.0084 | 0.0074 | 0.0025 | 0.0013 | 0.0005 | 0.0011 | 0.0003 | 0.0001 |

Source: Authors' calculations using data from the Human Mortality Database [23].

All coefficient $p$-values were less than $1 \times 10^{-6}$.

explain why the aforementioned approximately 0.02 unit decrease is the same for both life span disparity and life table entropy. This identification also helps us interpret the approximately 0.02 slopes in the Age 0 cases in Figs [1] and [2]. This interpretation is easiest to understand at the country-by-country level. For example, specializing for the moment to the Age 0 data for the French female population (Figs [1] (bottom) and [2] (bottom)), that $e''(e_0)/2 \approx 0.02 > 0$ suggests that the rate of change of remaining life expectancy evaluated at $e_0$ (that is, $e'(e_0)$) tends to increase over time in this population. Indeed, a closer look at the one-year change in $e'(e_0)$ in the population confirms this trend (see Fig 5 in Appendix B.4 of [S1 Appendix]. This suggests a flattening of the graph of the population's remaining life expectancy function $e(x)$ near $x = e_0$ in recent times; Fig 6 in Appendix B.4 of [S1 Appendix] illustrates this for French males and females.

Fig 7 in Appendix B.4 of [S1 Appendix] furnishes a more demographic interpretation of $e''(e_0) \approx 0.04$, which we extract as follows. First, we note that the Figure shows that remaining life expectancy decreases as age increases beyond 40 (i.e., $e'(x) < 0$ for $x \geq 40$). Next, we observe that these decreases are not uniform: older individuals lose less life expectancy per unit increase in their age than do younger individuals (thus $e'(x)$ is increasing with $x$, so that $e''(x) > 0$). This has the effect of *decreasing* the amount of remaining life expectancy that is lost with each unit increase in individuals' age. What the regression result of $e''(e_0) \approx 0.04$ says, then, is that if one compares these decreases across life tables with different $e_0$-values (as our regressions have done), more recent life tables have decreases that are approximately 0.04 years lower than more distant life tables. For example, in 2001, the approximate drop in remaining life expectancy resulting from aging 1 year from age 82.9 (the approximate $e_0$-value for that year) was about 0.62 years. By 2004, the approximate drop in remaining life expectancy resulting from aging 1 year from age 83.9 (the approximate $e_0$-value for that year) was about 0.57 years; this is about 0.05 years lower than the 2001 value. Crudely summarized, then, the finding that $e''(e_0) \approx 0.04$ indicates that, over time, the drop in the remaining life expectancy of French females surviving to age $x = e_0$ is itself dropping by approximately 0.04 years for each unit increase in $e_0$. French females, therefore, are not only living longer but enjoying less of a drop in remaining life expectancy when reaching age $x = e_0$ and surviving one more year.

In summary, these insights show that the curvature of the remaining life expectancy function—specifically, its degree of convexity at age $e_0$—drives most of the changes in life span disparity and life table entropy after a one-unit change in the variance and index of dispersion, respectively, associated with the underlying deaths distribution, with similar insights holding for the other $a$-values presented in Figs [1] (bottom) and [2] (bottom).

Returning now to Figs [1] (top) and [2] (top), one can ask if the same insights apply to the broader HMD data set. While we cannot, at this aggregated level, directly interpret the slopes in the regressions in Figs [1] (top) and [2] (top) in terms of the remaining life expectancy function and its higher-order derivatives—because the trends in remaining life expectancy vary across countries in the data set—nonetheless [24] and [25] imply that these slopes *are* associated with the convexity of the remaining life expectancy functions in each country in the data set. This is in fact what our results indicate. Regression equations fitted separately by country, sex, and age confirm a similar pattern in HMD countries as in the French population (Fig 8 in [S1 Appendix]). For example, all coefficient estimates that represent $e''(e_0)/2$ are positive and small in magnitude, suggesting a slow rate of decline in remaining life expectancy over time at age $e_0$. We note that these results also highlight important sex and country differences in the speed of change in remaining life expectancy; however, a detailed description of these results is beyond the scope of the current paper.

Next, let us illustrate the value of incorporating additional central moments into our analyses. We illustrate this by returning to [20] and [23], which incorporate up to three moments,

and studying the French female population. Tables 1 and 2 below show the results of regressing $e^{\dagger}(a)$ and $H(a)$ against progressively more central moments, up to $m_3(a)$ and $\widehat{m}_3(a)$, for that population.

We note first that the high adjusted $R^2$-values for the $\sigma^2$-only regressions imply that differences in the variance of the underlying deaths distributions explain most of the variation in $e^{\dagger}$- and $H$-values across all ages among females, but not among males at age 40 (see Tables 1 and 2 for females, and Tables 3 and 4 in S1 Appendix for males). As indicated by the adjusted $R^2$-values in both the $e^{\dagger}$ and $H$ regressions, each additional moment incorporated reduces the remaining unexplained variance and decreases the regression standard errors (last row in the Tables). Since incorporating additional moments into our analysis is equivalent to increasing the $k$-value in (13) and (14), these decreases in standard errors verify empirically what we proved in Theorem 1: that the error in approximating $e^{\dagger}(a)$ and $H(a)$ with their corresponding truncated series decreases as $k$ increases. Our empirical results suggest that knowing the first three central moments of the deaths distribution—the variance and, effectively, skewness and kurtosis—are sufficient to accurately estimate life span disparity and life table entropy values. Moreover, adding more moments to the regression greatly improves our understanding of the link between $e^{\dagger}$ and $H$ in relation to the underlying deaths distribution. For example, among males in middle-adulthood (age 40, Tables 3 and 4 in S1 Appendix), the adjusted $R^2$ changes from 0.4478 when only $\sigma^2$ is included in the model for $e^{\dagger}$ to 0.9951 when $m_3$ is added, and to 0.9971 when $m_4$ is also added to the model, and similarly for the $H$-regressions. That is, at age 40, life span disparity and life table entropy depend not only on the dispersion in ages at death (i.e., $\sigma^2$) but also on the shape of the death distribution in terms of its asymmetry (skewness) and less so on the tails of the distribution (kurtosis). For instance, for both females and males, the magnitude of the $m_3$ coefficient is progressively larger at older ages suggesting that as the death distribution at these ages becomes more asymmetric there would be increases in life span disparity and in the life table entropy. These insights exemplify how our results can be used to accurately forecast changes in life span disparity and life table entropy resulting from changes in the underlying deaths distribution. Similar conclusions to those presented above hold for the broader HMD data set for all $a$-values studied in Figs 1 and 2 (see Appendix B.3 in S1 Appendix, e.g., Tables 5 and 6 in S1 Appendix for females and males, respectively, for the HMD version of Table 1, and Tables 7 and 8 in S1 Appendix for females and males, respectively, for the HMD version of Table 2). Collectively, these results illustrate the additional information and insights that empirical studies of life span inequality, like the ones mentioned in the Introduction, miss by excluding higher-order moments of the deaths distribution, particularly in middle-adulthood and at older ages. Finally, we note once more that the coefficients of the moments in Tables 5–8 in S1 Appendix can be interpreted again in terms of the higher-order derivatives of the remaining life expectancy function, as we did previously in the context of (24)–(25).

These empirical studies show that the variation in the life span disparity and life table entropy values across the 41 countries in the HMD is largely explained by differences in the associated $\sigma^2(a)$ and $v_d(a)$ values (as evidenced by the high adjusted R-squared values in Tables 5–8 in S1 Appendix). In addition, our results show that in middle-adulthood, the variation in the life span disparity and life table entropy also depends on the shape of the deaths distribution in relation to the skewness (which is proportional to $m_3$) and kurtosis (which is proportional to $m_4$), particularly among males. But it is only because of the theoretical foundation provided by (20)–(23) that we can understand both the demographic origins of the coefficients in the regressions in Figs 1 and 2 and Figs 3,4 in S1 Appendix how those coefficients might change country-by-country and in response to changes in each country's underlying remaining life expectancy function.

## Discussion

In this paper we focused on building a bridge between well-known statistical and demographic descriptors of mortality. In particular, we centered our work on the deaths distribution and established exact relations between known demographic and statistical descriptors of mortality defined in terms of life table parameters. Our main theoretical results—Theorem 1 and Corollary 1.1—are mathematical theorems; they are exact relationships, not empirical regularities. As such, they provide the missing explicit connections between statistical and demographically-meaningful measures of variation in the life table deaths distribution in human populations that we discussed in the Introduction. These missing connections take the form of the expansions (13)–(16), which make it possible to understand, from a theoretical standpoint, exactly how life span disparity and life table entropy depend on the moments of the deaths distribution. These expansions show that $e^{\dagger}(a)$ and $H(a)$, two widely used metrics of life span inequality in demography developed with specific demographic applications in mind, can be decomposed into sums of products of classic statistical measures of dispersion (the $m_n(a)$) with measures of remaining life expectancy and its rates of change (the $e^{(n)}(e_a)$). These terms have straightforward demographic interpretations, and thus (13)–(16) offer insights into the levels and changes in life span inequality in populations based on analyses of their $m_n(a)$ and the $e(x)$ functions near $x = e_a$.

We first illustrated this new approach in the context of the hyperbolic mortality model, where we saw that the infinite series (15) and (16) collapsed to just their first terms. Though this is a rather special case, we point out that the series expansions all have the form "$e(e_a)$ plus additional terms." The significance of this is the revelation that both life span disparity and life table entropy—regardless of the associated $a$-value—are constructed from the remaining life expectancy function, with additional contributions coming from the second- and higher-order moments of the deaths distribution modulated by the higher-order derivatives of that same remaining life expectancy function. This is perhaps not surprising, given the original definitions (1), but the central role played by $e(e_a)$ and the particular interplay of the derivatives of $e(x)$ and the (central) moments of the deaths distribution is quite non-trivial.

Moreover, the expansions (20) and (22) generalize the study of how life span disparity and life table entropy depend on the moments of the deaths distribution at any age (what we called "$a$-truncated deaths distributions"). This approach allows us, for example, to quantify the importance of the shape of the death distribution in middle-adulthood and at older ages in assessing changes in life span disparity. For example, there is ample evidence documenting sex differences in mortality and survival in favor of females [28–30]. Our results from the HMD data for both $e^{\dagger}$ and $H$ also show this sex difference, with a much larger variation in these indicators at all ages among men than women. Further, our results reveal that in middle-adulthood (at age 40) the skewness of the deaths distribution plays a larger role among men than women in explaining changes in life span inequality (e.g., the adjusted $R^2$-value largely increases when $m_3$ and $\widehat{m}_3$ are included in the regressions for men but not for women). This suggests a more asymmetric distribution in ages at death at younger ages among men than women, indicating their higher underlying risk of death. This result is in line with previous evidence indicating that the narrowing sex differential in life expectancy in high-income populations can be explained "because women's deaths are less dispersed across age (i.e., survivorship is more rectangular)" [31].

From an empirical standpoint, our new connections have established specific correlations between well-known demographic and statistical descriptors of life span inequality and the central moments of the deaths distribution. These are summarized in the various regression results across Figs 1 and 2 and Figs 3,4 and Tables 5–8 in S1 Appendix, and provide ample and

robust empirical evidence for the associations predicted by (20)–(23). Our equations also make clear why there is a high correlation between $e^\dagger$, $H$ and various measures of dispersion in ages at death that have been empirically reported in the literature (e.g., the variance, standard deviation, etc.) [18]. For example, it has been postulated that: "Intuitively it appears that measures of rectangularity and variability should be related inversely: as the distribution of age at death becomes less variable, the survival curve should become increasingly rectangular" [18, pp 479]. Because a more rectangular survival curve leads to smaller values in $e^\dagger$ and $H$, this statement would suggest that reductions in the variance in ages at death (or the index of dispersion) should lead to smaller values in $e^\dagger$ and $H$. Our empirical results support this conclusion. More broadly, our theoretical and empirical results are consistent with the existing literature and provide additional insights into the underlying deaths distribution in studies that have focused on life table indicators such as life disparity and the modal age at death. For example, looking at 40 of the most developed countries during the period 1840–2009, countries with the highest life expectancy at birth also attained the lowest life span disparity [13]. Due to the robust relations we have documented herein, these findings imply a low variance (hence, low standard deviation) in the ages at death in those countries during that time period (via our $e^\dagger$ and $\sigma^2$ relations, Fig 1 (top) and Fig 3 in S1 Appendix (top)). Moreover, our results further elucidate that the shape of the deaths distribution also contributes to changes in life span disparity whereby more asymmetric distributions are linked with increases in life span disparity. These examples illustrate how our results can be combined with known empirical correlations to yield new insights into mortality trends.

We note also that our results contain potential policy implications. For example, let us return to the life span disparity regressions in Fig 1 and Fig 3 in S1 Appendix. The slopes of those regressions measure how fast life span disparity decreases following a one-unit decrease in the variance or index of dispersion in the deaths distribution. Greater slope values, therefore, would lead to greater declines in life span inequality (as measured by life span disparity). But absent a theoretical understanding of what drives those slopes values, policy makers would be unable to design interventions to increase those slope values. However, because of the revelation that those slopes are related to $e''(e_a)$, such interventions are now conceivable. Furthermore, the built-in $a$-dependency means that policymakers could target their interventions to address life span inequality trends at certain ages (e.g., ages 80+, for which, as we noted, life span inequality has in general been increasing since 1950).

Beyond the new results and insights described above, this article builds a bridge between the rich literature on life span disparity and life table entropy within demography on the one hand and the vast literature concerning probability density functions (like the deaths distribution) and their statistical properties on the other. This bridge has the potential to spark at least three profitable research programs. The first relates to Eqs (13) and (14). These equations connect the moments of the deaths distribution in robust and specific ways to demographically-meaningful descriptors of mortality. Thus, our work has the potential to provide statistical foundations for empirically-driven findings in demography that either pertain directly to $e^\dagger$ or $H$ or involve these parameters (e.g., [15]). Another potentially profitable research program concerns the time evolution of demographically-meaningful descriptors of mortality. Briefly, suppose we now allowed the deaths distribution to become time-dependent: $f(x, t)$ in our notation. Then Eqs (7) and (8) would acquire a time dependency. Owing to the relations we have established here between those moments of the deaths distribution and demographic measures like life span disparity and life table entropy (c.f., Theorem 1 and Corollary 1.1), those time-dependent moments would translate into time-dependent life table parameters (e.g., $e^\dagger(t)$). Thus, we would obtain an explicit connection between the temporal evolution of demographically-meaningful indicators of mortality and the deaths distribution itself. Were one to then

apply what is known to mathematicians and statisticians about time-dependent density dynamics to the deaths distribution, this line of investigation would allow one to extract insights into the evolution of mortality in human populations. We therefore advocate for future research to focus on understanding the (temporal) dynamics of the deaths distribution. Any insights emerging from such research would, via the time-dependent versions of (7) and (8) and the interconnections established in this article, lead to new insights into the temporal evolution of demographically-meaningful metrics like life expectancy, life span disparity, and related measures. These insights would complement the theoretical and empirical analyses reported herein and expand our understanding of the trajectory of mortality in human populations. Finally, because our theoretical results hold for a general age-structured population (subject to the hypotheses we have assumed), they are broadly applicable to not just human populations but to other age-structured populations as well. This makes possible the study of the same types of questions we have investigated herein—for example, the relation of life span disparity and life table entropy to the moments of the deaths distribution—for non-human populations. It also holds the potential to provide the same demographic and statistical insights into the dynamics of mortality, this time across the tree of life.

## Supporting information

**S1 Appendix.** A Mathematical Derivations [32]. B Supplementary Information [23]. (PDF)

## Acknowledgments

We thank the referee and the editor for reviewing the manuscript, and the production team for their assistance with the production process.

## Author Contributions

**Conceptualization:** Oscar E. Fernandez, Hiram Beltrán-Sánchez.

**Data curation:** Hiram Beltrán-Sánchez.

**Formal analysis:** Oscar E. Fernandez, Hiram Beltrán-Sánchez.

**Investigation:** Oscar E. Fernandez, Hiram Beltrán-Sánchez.

**Methodology:** Oscar E. Fernandez, Hiram Beltrán-Sánchez.

**Software:** Hiram Beltrán-Sánchez.

**Visualization:** Hiram Beltrán-Sánchez.

**Writing – original draft:** Oscar E. Fernandez, Hiram Beltrán-Sánchez.

**Writing – review & editing:** Oscar E. Fernandez, Hiram Beltrán-Sánchez.

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
