## [Decision Letter · Decision Letter 0]

7 Jan 2022

Life span inequality as a function of the moments of the deaths distribution: connections and insights.

PONE-D-21-33137

Dear Dr. Fernandez,

We’re pleased to inform you that your manuscript has been judged scientifically suitable for publication and will be formally accepted for publication once it meets all outstanding technical requirements.

Kind regards,

Srinivas Goli, Ph.D.

Academic Editor

PLOS ONE

Additional Editor Comments (optional):

Congratulation. I really enjoyed reading this paper.

Reviewers' comments:

Reviewer's Responses to Questions

**Comments to the Author**

1. Is the manuscript technically sound, and do the data support the conclusions?

Reviewer #1: Yes

2. Has the statistical analysis been performed appropriately and rigorously? 

Reviewer #1: Yes

3. Have the authors made all data underlying the findings in their manuscript fully available?

Reviewer #1: Yes

4. Is the manuscript presented in an intelligible fashion and written in standard English?

Reviewer #1: Yes

5. Review Comments to the Author

Reviewer #1: I like to congratulate authors for good work. Your empirical findings show that the survival curve is becoming more rectangular as the dispersion declines. It also explicitly links the temporal evolution of demographically-meaningful indicators of mortality and death distribution.

6. PLOS authors have the option to publish the peer review history of their article (what does this mean?). If published, this will include your full peer review and any attached files.

Reviewer #1: **Yes: **Sayeed Unisa

---

## [Editor Report · Acceptance letter]

21 Jan 2022

PONE-D-21-33137 

Life span inequality as a function of the moments of the deaths distribution: connections and insights 

Dear Dr. Fernandez:

I'm pleased to inform you that your manuscript has been deemed suitable for publication in PLOS ONE. Congratulations! Your manuscript is now with our production department. 

Kind regards, 

on behalf of

Dr. Srinivas Goli 

Academic Editor

PLOS ONE